# Beyond IC$_{50}$—A computational dynamic model of drug resistance in enzyme inhibition treatment

**J. Roadnight Sheehan**[1], **Astrid S. de Wijn**[1]\*, **Thales Souza Freire**[2], **Ran Friedman**[3]

**1** Department of Mechanical and Industrial Engineering, Norwegian University of Science and Technology, Trondheim, Norway, **2** Institute of Physics of the University of São Paulo, Department of General Physics, São Paulo, Brazil, **3** Department of Chemistry and Biomedical Sciences, Linnaeus University, Kalmar, Sweden

\* astrid.dewijn@ntnu.no

**Data Availability Statement:** Code and input data are available at https://github.com/JenWrenPen/BeyondIC50.

## Abstract

Resistance to therapy is a major clinical obstacle to treatment of cancer and communicable diseases. Drug selection in treatment of patients where the disease is showing resistance to therapy is often guided by IC$_{50}$ or fold-IC$_{50}$ values. In this work, through a model of the treatment of chronic myeloid leukaemia (CML), we contest using fold-IC$_{50}$ values as a guide for treatment selection. CML is a blood cancer that is treated with Abl1 inhibitors, and is often seen as a model for targeted therapy and drug resistance. Resistance to the first-line treatment occurs in approximately one in four patients. The most common cause of resistance is mutations in the Abl1 enzyme. Different mutant Abl1 enzymes show resistance to different Abl1 inhibitors and the mechanisms that lead to resistance for various mutation and inhibitor combinations are not fully known, making the selection of Abl1 inhibitors for treatment a difficult task. We developed a model based on information of catalysis, inhibition and pharmacokinetics, and applied it to study the effect of three Abl1 inhibitors on mutants of the Abl1 enzyme. From this model, we show that the relative decrease of product formation rate (defined in this work as "inhibitory reduction prowess") is a better indicator of resistance than an examination of the size of the product formation rate or fold-IC$_{50}$ values for the mutant. We also examine current ideas and practices that guide treatment choice and suggest a new parameter for selecting treatments that could increase the efficacy and thus have a positive impact on patient outcomes.

## Author summary

Resistance to drug treatments is a major problem in treating diseases. Choosing the right drug for a patient who shows resistance to treatment is a tricky business. Using a model of the treatment of chronic myeloid leukaemia (CML), we challenge the current methods of using IC50 values to guide treatment selection. CML is a blood cancer that is treated with drugs called Abl1 inhibitors. Around one in four patients will show resistance to the first treatment they are given. The Abl1 inhibitors target the Abl1 enzyme, mutations in this

**Funding:** The author(s) received no specific funding for this work.

**Competing interests:** The authors have declared that no competing interests exist.

enzyme are the most common cause for drug resistance. Different drugs work better for different mutations, but how this happens is not fully understood, which makes choosing treatment difficult. We developed a model based on how the Abl1 enzyme works, how the inhibitor affects the enzyme, and drug concentrations in patients. The model shows that the relative decrease in the efficacy of the enzyme (which we named "inhibitory reduction prowess") better indicates which drugs should be selected for which mutation than the usual current methods. We also examine other factors that guide treatment choice and suggest a new method for selecting treatments that could have a positive impact on patients.

## 1 Introduction

Contemporary drug design relies on the identification of molecular targets, such as receptors or enzymes, that once activated or inhibited lead to a marked improvement in the disease state. One caveat of this paradigm is the risk for development of resistance where the disease to be treated is either cancer or a communicable (transmittable, pathogen borne) disease. Chronic myeloid leukaemia (CML) has been extensively studied as a model for the development of drug resistance especially when it comes to resistance due to mutations in the drug target [1, 2].

CML is a blood cancer that in the vast majority of cases (95% [3]) involves a chromosomal translocation (the Philadelphia chromosome), which can be detected and is the basis for diagnosis. The chromosomal translocation results in a fusion of the breakpoint cluster region (BCR) of chromosome 22 with the Ableson leukemia (Abl) gene on chromosome 9. This BCR-Abl region encodes Abl1 that lacks its regulatory part. The deregulation promotes overactivity of the translated protein, Abl1 kinase, and this in turn disrupts cells' processes, leading to: increased growth and multiplication; blocking of differentiation; and loss of programmed cell death [4]. The Philadelphia chromosome is not only the primary cause for CML, but also a contributing factor to other cancers [5].

Abl1 inhibitors, are the primary treatment for adults with CML [6]. The first Abl1 inhibitor treatment that is most commonly given to patients is imatinib, which has milder side effects than other Abl1 inhibitors [7]. Imatinib and other Abl1 inhibitors reduce cell proliferation, eliminating the symptoms of CML and rendering the presence of Philadelphia chromosomes undetectable [8]. However, treatment with Abl1 inhibitors does not provide a cure for the disease and patients often need treatment for the rest of their lives [9].

Although Abl1 inhibitor treatment is increasing survival rates, resistance to imatinib as initial treatment develops in around 25% of patients within two years [10] (estimations for other Abl1 inhibitors are: 20% for initial treatment with dasatinib within two years [11]; and 21% for secondary treatment after treatment of one prior tyrosine kinase inhibitor within one year [12]). In most cases, the resistance arises from mutations in the Abl kinase domain that alter the protein structure. If an Abl1 mutant is less affected by the Abl1 inhibitor, cells with this mutation then become prevalent in the cell population, and increase the growth rate [13]. Table 1 gives information about the Abl1 inhibitors and the mutations that are associated with resistance to those treatments that we examine in this work.

Once significant cell populations with imatinib resistant mutations arise, the treatment must change. Second- and third-generation Abl1 inhibitor treatments are available. However, which drug should be matched to which mutation is not always clear, and overall, there is a need to develop treatment protocols that will minimise the risk for resistance [15, 16], which

**Table 1. The table (refined data from [14]) shows the mutants associated with resistance for imatinib, ponatinib, and dasatinib.** These mutants were considered in this work as they are the most common and have been extensively studied.

| Drug | imatinib | ponatinib | dasatinib |
|---|---|---|---|
| **Mutants considered in this work** | G250E<br>E255K<br>E255V<br>T315I<br>T315M<br>Y253H | E255K*<br>E255V*<br>T315I*<br>T315M | T315I<br>T315M |

* Although these mutations do have an increased IC$_{50}$ for ponatinib, they do not lead to resistance alone, but do so as compound mutations.

requires an understanding of resistance mechanisms that take into account the effects of drug binding, enzyme activation [17, 18] and interference with catalysis [19]. The clinical toxicity (and financial cost) of these inhibitors in long term use is a strong motivator to achieve higher rates of treatment free remission [20–23], especially in younger patients [20]. This is potentially achievable after long periods (>2 years [22, 23]) of deep molecular response to treatment. For the best chance of achieving this deep molecular response, the ability to accurately choose treatments is of high importance.

As with many other diseases, treatment of resistance to therapy with patients with CML is currently guided by IC$_{50}$ values. However, the reliability of IC$_{50}$ values can be contested [24]. In vitro measurements of IC$_{50}$ can vary greatly between different assays, revealing possible inaccuracies, and therefore decreased reliability, in the data [25, 26]. Collecting IC$_{50}$ values from different sources can increase the noise of the data presented [27]. Furthermore, the relationship between product formation rate and cell growth rate is not necessarily linear, adding further noise to comparisons or models deriving their values from such data [28].

In this paper, we develop a computational model to assess whether the product formation rate of the Abl1 enzyme and its mutations provides an indicator of resistance to Abl1 inhibitor treatment. We thereafter compare the leading approach of drug selection in CML treatment (relative IC$_{50}$, see section 2) with methods that include further information (such as Abl1 inhibitor concentration data) to assess their ability to infer drug resistance, which would better inform choices in treatment when resistant mutations arise.

## 2 Approach to understanding resistance

Many factors have been examined in the pursuit of finding the best choice of Abl1 inhibitor to switch to once resistant mutations arise [19, 29–31]. Often the data of the wild-type (WT) Abl1 enzyme is compared to that of the mutation (Mut)—specifically, the IC$_{50}$ values. IC$_{50}$ is the concentration of the inhibitor in which the enzyme's activity or cell's growth (depending on the assay) reduces to half of its maximum value. Typically in drug selection, an examination of the relative IC$_{50}$ of each Abl1 mutation is used: a ratio of IC$_{50}^{Mut}$/IC$_{50}^{WT}$. This is often described as an imperfect approach [29, 31–33]. Generally, the lower the value of the IC$_{50}$, the greater the effect of the inhibitor. However, IC$_{50}^{Mut}$/IC$_{50}^{WT}$ quantifies which mutations each drug is most effective against, rather than which drug is best suited to treating each mutation. Furthermore, there are other factors that are relevant for choosing the treatments. A patient's tolerance and the safety profile of the treatments must be considered as well [22, 23].

An additional complication for the selection process is that the mutations that give rise to resistant Abl1 variants affect other properties than drug binding. For example, the catalytic rate constant, $k_{cat}$, varies between mutations, indicating that different mutations may provide a

higher product formation rate. However, due to other unknown factors (such as whether the active state or inactive state is more stable), it is unwise to assume that the size of $k_{cat}$ is solely responsible for (or has any significant effect on) whether a mutation is resistant to a certain treatment.

Another method of examining and comparing Abl1 with its mutations is catalytic efficiency:

$$\frac{k_{cat}}{K_M} = \frac{k_{cat}}{k_{cat} + k_{off}^S} k_{on}^S \, , \tag{1}$$

where $K_M$ is the Michaelis constant for the enzyme; $k_{off}^S$ is the rate constant of the substrate unbinding without catalysis from the enzyme; and $k_{on}^S$ is the rate constant of the substrate binding to the enzyme per unit concentration of the substrate. A compound mutation is a double mutation in the same allele (rather than one mutation in one allele and another mutation in another allele). In cases where two mutations do not show resistance alone, but do so as a compound mutation, the compound mutation might have higher catalytic efficiency than either of the single mutations [19]. This highlights the possibility that the catalytic efficiency might affect the resistance, and therefore we consider the catalytic efficiency in our investigations.

In this paper, we take a mathematical, computational approach to investigate the change in the product formation rate of the wild-type (WT) BCR-Abl Abl1 enzyme and six of its mutations when subjected to treatment with one of three different Abl1 inhibitors. Our aim is to provide insight into how these changes in the product formation rate lead to drug resistance. We also investigate analytically the link between a drug and mutation combination showing resistance and different measurable quantities of the mutant Abl1 enzymes (such as $k_{cat}$, $K_M$ and IC$_{50}$).

The Abl1 inhibitors in focus in the work are imatinib, ponatinib and dasatinib. With the computational model, we examine these inhibitors in combination with the WT Abl1 enzyme and six mutations of it: G250E, E255K, E255V, T315I, T315M, and the compound mutation Y253H-E255V. When analytically investigating different possible indicators of resistance, we also examine the mutations Y253H, G250E-T315I, Y253H-T315I, and E255V-T315I.

## 3 Methods

The success of Abl1 inhibitors in the clinic shows that enzyme inhibition is coupled to a reduction in the product formation rate of Abl1. In a clinical setting, treatment is monitored through hematologic (peripheral blood counts of leukocytes, platelets, and immature cells), cytogenetic (the prevalence of the Philadelphia chromosome in cells), and other molecular measures of blood or bone marrow samples from patients [34]. These tests are population counts and considering the Abl1 enzyme as a key part of growth and multiplication signalling, this might suggest that the product rate is a good indicator of resistance. Consequently, we use computational modelling to describe the product rate for different drug resistant mutations under different treatments. Based on our model, we analytically derive an indicator of the severity of the resistance in drug-mutation combinations that can be used in practical settings to better inform treatment choice.

### 3.1 Modelling of drug resistant mutations

To establish a dynamic model, we consider the different states that a system consisting of an enzyme, substrate, and inhibitor can be in and derive a complete set of time-dependent rate

equations for the concentrations of enzyme in each state,

$$\frac{\mathrm{d}[\mathrm{E}_x](t)}{\mathrm{d}t} = -[\mathrm{E}_x](t) \sum_y \kappa_{x \to y} + \sum_y [\mathrm{E}_y](t) \kappa_{y \to x} \ , \tag{2}$$

where $x$ and $y$ are the labels of the enzyme states. We denote the concentration of the enzyme in state $\mathrm{E}_x$ by $[\mathrm{E}_x]$, and $\kappa_{x \to y}$ is the rate constant for transition from state $\mathrm{E}_x$ to state $\mathrm{E}_y$.

To set up this model, we need two components: the states that need to be considered, and the transition rates between them. In section 3.1.1, we determine the relevant states of the system. Thereafter, we examine the system by assuming quasi-equilibrium conditions in section 3.1.2. Based on this, we then derive the relationships between the rate constants. Quantitative values and approximations for the rate constants can then be made based on existing experimental results and simulations, which we describe in section 3.1.3.

To more accurately model the physiology of drug treatment, where the concentration fluctuates, we also consider the pharmacokinetics of the inhibitor in the system as described in section 3.1.4. In section 3.1.5, we outline the calculations of the initial conditions of the system analytically, assuming a pre-existing quasi-equilibrium steady state before treatment.

The mass balance equations for the model can be found in Tables A and B in S1 Text.

**3.1.1 Determining the states of the system.** As the Abl1 enzyme is part of a signalling pathway that controls cellular growth, its deregulation results in rampant cell proliferation. The cellular system is complex, but as the Abl1 signalling is of high importance in the pathway, the system can be simplified to focus on one element of CML in patients—the phosphorylation of tyrosine residues in various proteins by Abl1, i.e. the product formation rate. The importance of this mechanism in CML is verified by its successful treatment with Abl1 inhibitors [35]. Focusing on the product formation rate as a measure for the efficacy of the drugs allows us to quantify and compare the many drug-resistant Abl1 enzymes with small changes in the BCR-Abl gene.

We first examine the possible states that the system can be in. The enzyme can be active or inactive. Moreover, it can be bound or not to the inhibitor, the substrate protein, and ATP. There are many combinations of these that could be considered. However, we can eliminate some of these combinations from our simplified computational model. Firstly, the site that the inhibitory drugs bind to is the same site that the ATP molecule binds to, therefore the ATP and an inhibitor cannot be bound to an Ab1 enzyme at the same time. Furthermore, if bound to an inhibitor drug molecule, the flexibility of the protein structure of the enzyme is reduced and prevents the binding of the substrate protein to Abl1 [36]. Thus, a combination of the inhibitor and the substrate protein binding with the enzyme together does not need to be considered in our model. The likelihood of the substrate binding to the inactive form of the Abl1 is very small [37]; we therefore do not consider the states with binding of the substrate to the inactive enzyme in our model. We also combine multiple transitions into single transitions between states (i.e. active unbound state to active bound to both the substrate protein and ATP, without transition via binding to one of the molecules). This leaves a total of five different states for the model: active enzyme not bound to anything; inactive enzyme not bound to anything; active enzyme bound to both the substrate protein and an ATP molecule; active enzyme bound to an inhibitor; and inactive enzyme bound to an inhibitor, as shown in Fig 1. Fig 1 also shows the transition rate constants between states, which will vary depending on the Abl1 inhibitor and mutation in the system and are discussed in detail in subsection 3.1.3.

Finally, we can reduce the number of possible states further by considering the binding of the different inhibitor drug molecules to the enzyme. Imatinib and ponatinib both bind only

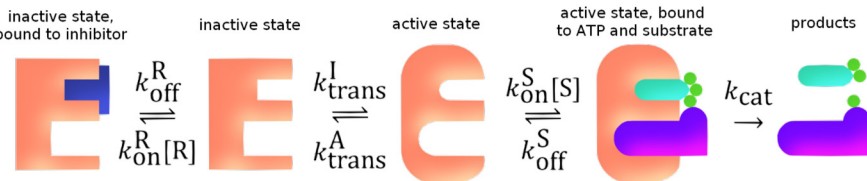

A    The model with imatinib and ponatinib.

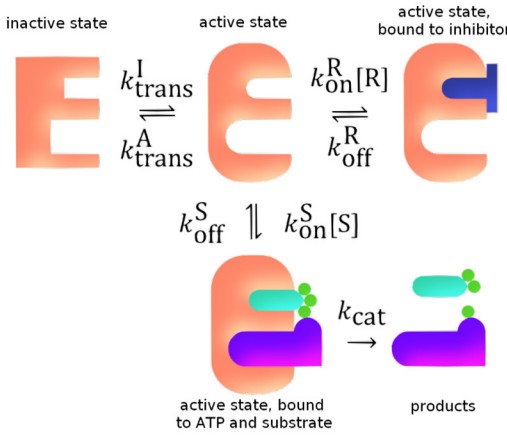

B    The model with dasatinib.

**Fig 1. A cartoon of the simplified model of the Abl1 enzyme system.** The states, molecules they bind to, the transitions between states, and the rate constants for each transition are shown. The Abl1 enzyme is represented by the peach-coloured E-shape in both its active (round corners) and inactive (sharp corners) states; the inhibitor is illustrated by the dark blue T-shape (A: either imatinib or ponatinib, B: dasatinib); and the remaining shapes depict the ATP with its phosphate groups and the protein substrate that is phosphorylated. Transition arrows are accompanied by their rate constants. The rates between states will differ depending on the mutation and Abl1 inhibitor present in the system.

to the inactive enzyme state while dasatinib binds to the active enzyme state [38, 39]. This leaves just four states of the Abl1 in each simulation.

**3.1.2 Fixed conditions and quasi-equilibrium.**    In order to develop our model for the system dynamics with fluctuating inhibitor concentration, it is useful to first consider the system with fixed inhibitor concentrations and assume a quasi-equilibrium. This system can be used to relate the various rate constants to one another, and can serve as a basis to construct equations for the dynamics.

Under fixed conditions, the system behaves in a relatively straight-forward way that can be determined using experimental and simulation data. We denote the concentration of substrate with [S] and the concentration of inhibitor with [R]. We assume that ATP is in surplus relative to the substrate and its concentration is therefore not treated explicitly by the model. The free-energy difference between the active and inactive enzymes states is denoted as $\Delta G^{I \to A}$, which is defined as $\epsilon_A - \epsilon_I$ where $\epsilon_{A,I}$ are the free energies of the active and inactive state of the enzyme, respectively. This value also describes the enzyme's state preference, i.e. whether the active or inactive state is more stable. Using principles of chemical kinetics [40] and statistical mechanics, we can find the proportions of the enzyme concentration that correspond to each of the states in quasi-equilibrium described in Fig 1. These proportions of the enzyme concentration are referred to as relative weights and are shown in Fig 2. The probability of finding an enzyme in a particular state is the weight of that state divided by the sum of all the weights

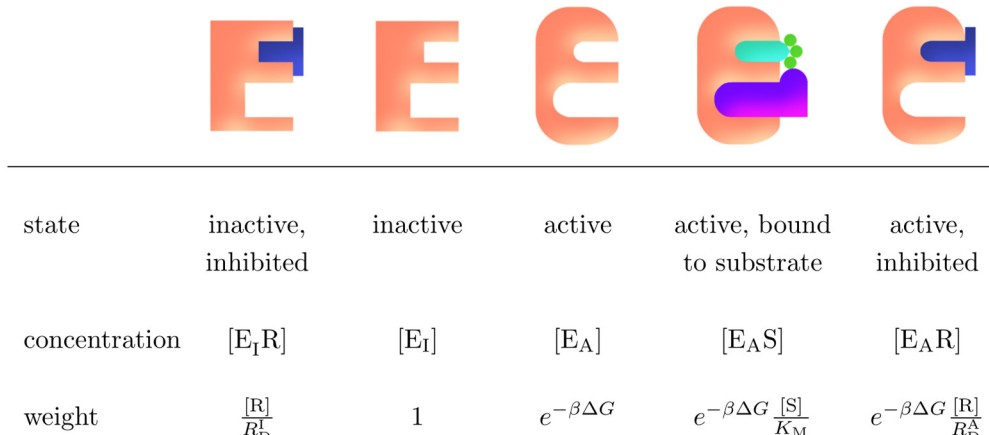

| state | inactive, inhibited | inactive | active | active, bound to substrate | active, inhibited |
|---|---|---|---|---|---|
| concentration | $[\text{E}_\text{I}\text{R}]$ | $[\text{E}_\text{I}]$ | $[\text{E}_\text{A}]$ | $[\text{E}_\text{A}\text{S}]$ | $[\text{E}_\text{A}\text{R}]$ |
| weight | $\frac{[\text{R}]}{R_\text{D}^\text{I}}$ | $1$ | $e^{-\beta\Delta G}$ | $e^{-\beta\Delta G}\frac{[\text{S}]}{K_\text{M}}$ | $e^{-\beta\Delta G}\frac{[\text{R}]}{R_\text{D}^\text{A}}$ |

**Fig 2. The enzyme states and their relative weights within the system in quasi-equilibrium.** [R] is the concentration of the inhibitory drug in the system; $R_\text{D}^\text{A,I}$ is the dissociation constant of the binding of the inhibitory drug for the enzyme in either active (A) or inactive (I) state; $\beta$ is the reciprocal of the product of Boltzmann's constant and the temperature ($\beta = 1/k_B T$); $\Delta G$ is the change in Gibbs free energy between the active and inactive enzymes, and it describes the preference of the unbound enzyme between the active and inactive states; [S] is the concentration of the substrates (it is assumed that ATP is in surplus relative to the substrate and its concentration is therefore not treated explicitly by the model); and $K_\text{M}$ is the Michaelis constant of the binding of the substrate to the active enzyme state.

present in the system. In this way, the total weight of this system is the partition function of a single enzyme. Note that in this work, we define $\beta$ as the reciprocal of the product of Boltzmann's constant and the temperature ($\beta = 1/k_B T$), instead of using the reciprocal of the product of the molar gas constant and the temperature ($1/RT$).

We can now express the product formation rate, $\frac{\text{d}[\text{P}]}{\text{d}t}$, for this system in terms of the concentration of active enzyme bound to the substrate $[\text{E}_\text{A}\text{S}]$, as in [40]:

$$\frac{\text{d}[\text{P}]}{\text{d}t} = k_\text{cat}[\text{E}_\text{A}\text{S}] \ , \tag{3}$$

where $k_\text{cat}$ is the turnover number (also known as the catalytic rate constant) for the active enzyme. The concentration of active enzyme bound to the substrate can be expressed as a proportion of the total concentration of enzymes, $[\text{E}_\text{tot}]$, in quasi-equilibrium. This is done using the weight for the state $[\text{E}_\text{A}\text{S}]$ from Fig 2 as a fraction of the total if the weightings for the system (which differs depending on the inhibitor present). Thus giving the product formation rate as:

$$\frac{\text{d}[\text{P}]}{\text{d}t} = k_\text{cat}[\text{E}_\text{tot}] \frac{e^{-\beta\Delta G}[\text{S}]/K_\text{M}}{W_\text{tot}}, \tag{4}$$

where the total weight, $W_\text{tot}$, is different depending on whether an inhibitor is present, and which inhibitor that is. For imatinib (i) and ponatinib (p) which bind to the inactive state of the enzyme (see Fig 1A) the total weight is:

$$W_\text{tot}^\text{i,p} = e^{-\beta\Delta G}\left(1 + \frac{[\text{S}]}{K_\text{M}}\right) + \left(1 + \frac{[\text{R}]}{R_\text{D}^\text{I}}\right) \ ; \tag{5}$$

and for dasatinib (d) which binds to the active state of the enzyme (see Fig 1B) the total weight

is:

$$W_{\mathrm{tot}}^{\mathrm{d}} = \mathrm{e}^{-\beta\Delta G}\left(1 + \frac{[\mathrm{S}]}{K_{\mathrm{M}}} + \frac{[\mathrm{R}]}{R_{\mathrm{D}}^{\mathrm{A}}}\right) + 1 \; . \tag{6}$$

$R_{\mathrm{D}}^{\mathrm{I}}$ and $R_{\mathrm{D}}^{\mathrm{A}}$ are the inhibitor dissociation constants.

**3.1.3 Quantitative determination of rate constants.** The real inhibitor concentration in patients varies with time, which in turn leads to fluctuations in the product formation rate, and potentially, a shift in the average. A more detailed discussion on and calculation of the fluctuations in the inhibitor concentration can be found in section 3.1.4. To take this into account, we must model the time-dependence of the concentrations of all the states and thus describe not only the quasi-equilibrium relative weights, but also the transition rates between states. These are modelled by the rate constants ($\kappa_{x\to y}$ and $\kappa_{y\to x}$) defined in Eq 2. In our model we calculate the rate constants as depicted in Fig 1 and expanded in Table 2. Our approach to obtaining these rate constants is described below.

**Abl1 inhibitor binding rate constants: $k_{\mathrm{off}}^{\mathrm{R}}$ and $k_{\mathrm{on}}^{\mathrm{R}}$.** To obtain $k_{\mathrm{off}}^{\mathrm{R}}$ we can make use of the experimentally determined inhibitor residence times ($t_{\mathrm{R}}$) from [41]. The residence time is a measure of the time the molecule is bound until its unbinding. The number of molecules unbinding from a bound state that occur per unit of time gives an unbinding rate, i.e. $^{1}/_{t_{\mathrm{R}}}$. From this we can obtain the $k_{\mathrm{off}}^{\mathrm{R}}$ for each Abl1 inhibitor for the WT:

$$k_{\mathrm{off}}^{\mathrm{R}} = \frac{1}{t_{\mathrm{R}}} \; . \tag{7}$$

These values can be found in Table 3.

For the rate constant in the other direction, $k_{\mathrm{on}}^{\mathrm{R}}$, we will make use of the quasi-equilibrium between the two rates, through the general definition of the dissociation constant of the inhibitor given as:

$$R_{\mathrm{D}} = \frac{k_{\mathrm{off}}^{\mathrm{R}}}{k_{\mathrm{on}}^{\mathrm{R}}} \; , \tag{8}$$

where $k_{\mathrm{off}}^{\mathrm{R}}$ is the rate constant of the inhibitor unbinding from the enzyme (measured in min$^{-1}$) and $k_{\mathrm{on}}^{\mathrm{R}}$ is the rate constant of the inhibitor binding per unit concentration of the inhibitor (usually measured in $\mu$M$^{-1}$min$^{-1}$).

**Table 2. To bridge two naming conventions (one more suitable to mathematical notation and one more conventional in enzymology), the table matches the rate constants as described in Eq 2 to those labelled in Fig 1.**

| rate constant of transitions as in Eq 2 | | rate constant as labelled in Fig 1 |
|---|---|---|
| $\kappa[\mathrm{E_I R}] \to [\mathrm{E_I}]$ | = | $k_{\mathrm{off}}^{\mathrm{R}}$ |
| $\kappa[\mathrm{E_I}] \to [\mathrm{E_I R}]$ | = | $k_{\mathrm{on}}^{\mathrm{R}}[\mathrm{R}]$ |
| $\kappa[\mathrm{E_I}] \to [\mathrm{E_A}]$ | = | $k_{\mathrm{trans}}^{\mathrm{I}}$ |
| $\kappa[\mathrm{E_A}] \to [\mathrm{E_I}]$ | = | $k_{\mathrm{trans}}^{\mathrm{A}}$ |
| $\kappa[\mathrm{E_A}] \to [\mathrm{E_A R}]$ | = | $k_{\mathrm{on}}^{\mathrm{R}}[\mathrm{R}]$ |
| $\kappa[\mathrm{E_A R}] \to [\mathrm{E_A}]$ | = | $k_{\mathrm{off}}^{\mathrm{R}}$ |
| $\kappa[\mathrm{E_A}] \to [\mathrm{E_A S}]$ | = | $k_{\mathrm{on}}^{\mathrm{S}}[\mathrm{S}]$ |
| $\kappa[\mathrm{E_A S}] \to [\mathrm{E_A}]$ | = | $k_{\mathrm{off}}^{\mathrm{S}}$ |
| $\kappa[\mathrm{E_A S}] \to [\mathrm{E_A}] + \text{Products}$ | = | $k_{\mathrm{cat}}$ |

**Table 3. Residence times, $t_R$, of the inhibitor molecules unbinding from the wild-type from [41] and the rates of $k_{off}^R$ calculated from them.**

| ABL1 inhibitor | $t_R$ (min) | $k_{off}^R$ (min$^{-1}$) |
|---|---|---|
| imatinib | 17 | 0.059 |
| ponatinib | 205 | 0.00488 |
| dasatinib | 249 | 0.00233 |

To obtain the dissociation constant, we use an approach similar to [40], making use of the experimentally measurable IC$_{50}$ value of the Abl1 inhibitor. We compare the product formation rate in Eq 4 for [R] = 0 and [R] = IC$_{50}$. As the IC$_{50}$ is defined as the concentration of inhibitor needed to halve the product formation rate, we can write:

$$\frac{d[P]}{dt}\bigg|_{[R]=IC_{50}} = \frac{1}{2}\frac{d[P]}{dt}\bigg|_{[R]=0.} \tag{9}$$

Substituting [R] = IC$_{50}$ into Eqs 5 and 6 and re-evaluating Eq 4, yields expressions for the inhibitor dissociation constants for the active-binding case, $R_D^A$, and the inactive-binding case, $R_D^I$:

$$R_D^A = \frac{IC_{50}e^{-\beta\Delta G}}{\eta + 1}, \tag{10}$$

$$R_D^I = \frac{IC_{50}}{\eta + 1}. \tag{11}$$

Here $\eta$ is used as short hand for

$$\eta = e^{-\beta\Delta G}\left(1 + \frac{[S]}{K_M}\right). \tag{12}$$

The IC$_{50}$ is experimentally known [42, 43]. We obtain a value for $\Delta G$ from molecular simulations, which are discussed in detail later when examining the rate of transition between the active and inactive states. The total substrate concentration [S] is chosen to be constant (i.e. an assumed perfect replenishment rate) and estimated to be 10 $\mu$M. Table 4 shows the $K_M$ values used in this work, obtained from [19], and the IC$_{50}$ values, from [42] and [43].

We can use these $k_{off}^R$ values for the WT; Eqs 10 and 11 for the dissociation constants; and Eq 8 to calculate the $k_{on}^R$ values for the WT. These inhibitor associated rate constants for the WT can be used to estimate the rate constants of the mutants based on what we know of the working mechanisms of the inhibitors. In order to make the simulations tractable, we assume that for the inactive-binding Abl1 inhibitors (imatinib and ponatinib), the $k_{off}^R$ remains the same across the WT and the mutants. This assumption is bolstered by a study that has shown that, once bound, a drug interacts with resistance associated mutants of a kinase in a similar manner to its interaction with the wild-type enzyme [44]. Similarly, for active-binding Abl1 inhibitors (dasatinib), we can assume the $k_{on}^R$ is the same across the WT and mutants where the mutation is not at the ATP binding site (G250E, E255K, E255V and Y253H). This assumption is made as the binding to the enzyme in the active state is determined by the structure of the ATP binding site in the active state. The same assumption does not hold for mutations in residues that form or border the ATP binding site (such as T315I and T315M). For such mutations, the $k_{on}^R$ values will be assumed to be different from the WT. To clarify, in this treatment

**Table 4. The values for $k_{cat}$ and $K_M$ are sourced from [19].** All the IC$_{50}$ values for the wild-type, G250E, E255K, E255V and T315I are from [42]. The IC$_{50}$ for imatinib for Y253H is also from this source. The other IC$_{50}$ values for Y253H along with all IC$_{50}$ values for T315M, G250E-T315I, Y253H-E255V, Y253H-T315I, and E255V-T315I are from [43].

| Mutation | $k_{cat}$ (min$^{-1}$) | $K_M$ ($\mu$M) | IC$_{50}$(nM) | | |
|---|---|---|---|---|---|
| | | | Imatinib | Ponatinib | Dasatinib |
| Wild-type | 66 | 17 | 527.0 | 2.1 | 1.8 |
| G250E | 175.2 | 14.3 | 3613.0 | 12.5 | 8.1 |
| Y253H | 26.9 | 4.6 | 4589 | 29.8 | 5.9 |
| E255K | 63.6 | 15.6 | 3174 | 17.6 | 10.3 |
| E255V | 6.8 | 22.1 | 8953 | 27.2 | 6.3 |
| T315I | 12.2 | 7.2 | 9221 | 6.3 | 137.3 |
| T315M | 8.1 | 1.9 | 10240 | 577.5 | 768 |
| G250E-T315I | 30.0 | 1.0 | 10240 | 152.4 | 768 |
| Y253H-E255V | 13.2 | 0.7 | 10240 | 203.5 | 18.1 |
| Y253H-T315I | 3.8 | 0.5 | 10240 | 357.9 | 768 |
| E255V-T315I | 4.1 | 0.6 | 10240 | 659.5 | 768 |

we reason that the binding of dasatinib is affected in a similar way to the binding of ATP, and the change in the substrate binding rate constant ($k_{on}^S$) from the WT reflects the change in binding of ATP for these mutants. Therefore, the change in $k_{on}^S$ from the WT to the mutant (which is detailed in the following section) is used to scale the WT values of $k_{on}^R$ to create suitable values for the T315I and T315M mutations with dasatinib. This also strengthens consistency within the model. The resulting values for the dissociation constants as well as for the binding and unbinding rate constants for the Abl1 inhibitors can be seen in Table 5.

**Substrate binding: $k_{off}^S$ and $k_{on}^S$.** The rate constants for binding and unbinding of the substrate to the enzyme can be estimated from experimental results of the product formation rate, in this case the coefficients $k_{cat}$ and $K_M$ for the WT and each mutation (Table 4). As we know most about the behaviour of the WT, we will use this as a starting point to compare the behaviour of the mutants to. In the following, we use superscript to indicate the WT or mutant (Mut). For the WT, $k_{cat}^{WT} = 66$ min$^{-1}$ [19]. With the WT we expect the enzyme to be more likely to catalyse than unbind, so that $k_{off}^{S,WT} < k_{cat}^{WT}$, while the rate constants should still be of the same order of magnitude. We therefore estimate $k_{off}^{S,WT}$ by $k_{cat}^{WT}/2$, i.e. 33 min$^{-1}$. Using the

**Table 5. The values calculated for the inhibitor dissociation constant, as detailed in Eqs 10 and 11 and approximated values for the binding rate constants of the inhibitors.**

| Mutation | $R_D$(nM) | | | $k_{off}^R$($\times 10^{-3}$min$^{-1}$) | | | $k_{on}^R$($\times 10^{-3}$nM$^{-1}$min$^{-1}$) | | |
|---|---|---|---|---|---|---|---|---|---|
| | imatinib | ponatinib | dasatinib | imatinib | ponatinib | dasatinib | imatinib | ponatinib | dasatinib |
| Wild-type | 203.6 | 0.811 | 0.695 | 59.0 | 4.88 | 2.33 | 6.01 | 0.290 | 3.35 |
| G250E | 1338 | 4.63 | 3.00 | 59.0 | 4.88 | 10.1 | 1.05 | 0.0441 | 3.35 |
| E255K | 12021 | 6.66 | 3.90 | 59.0 | 4.88 | 13.1 | 0.732 | 0.0491 | 3.35 |
| E255V | 3651 | 11.1 | 2.57 | 59.0 | 4.88 | 8.61 | 0.440 | 0.0162 | 3.35 |
| T315I | 2721 | 1.86 | 40.5 | 59.0 | 4.88 | 136 | 2.63 | 0.0217 | 3.35 |
| T315M | 1410 | 79.5 | 106 | 59.0 | 4.88 | 779 | 0.0614 | 0.0419 | 7.37 |
| Y253H-E255V | 628.8 | 12.5 | 1.11 | 59.0 | 4.88 | 3.72 | 0.391 | 0.0938 | 3.35 |

relationship:

$$K_{\mathrm{M}} = \frac{k_{\mathrm{cat}} + k_{\mathrm{off}}^{\mathrm{S}}}{k_{\mathrm{on}}^{\mathrm{S}}} \ , \tag{13}$$

we can calculate a value for $k_{\mathrm{on}}^{\mathrm{S,WT}} = 5.82 \, \mu\mathrm{M}^{-1}\mathrm{min}^{-1}$.

For the mutants, which in the absence of inhibitor are less fit than the WT, we expect the opposite, that the enzyme is more likely to unbind than to catalyze—i.e. $k_{\mathrm{off}}^{\mathrm{S,Mut}} > k_{\mathrm{cat}}^{\mathrm{Mut}}$, while the rate constants should again be of the same order of magnitude. We therefore use as a first estimate $k_{\mathrm{off}}^{\mathrm{S,Mut}} = 2k_{\mathrm{cat}}^{\mathrm{Mut}}$, which means that $k_{\mathrm{on}}^{\mathrm{S,Mut}} = 3k_{\mathrm{cat}}^{\mathrm{Mut}}/K_{\mathrm{M}}^{\mathrm{Mut}}$. However, the mutants should also not be so unfit as to not be able to compete at all with the WT when there is no treatment, meaning that $k_{\mathrm{on}}^{\mathrm{S,Mut}} \geq k_{\mathrm{on}}^{\mathrm{S, WT}}$. In this case $k_{\mathrm{on}}^{\mathrm{S,Mut}} = k_{\mathrm{on}}^{\mathrm{S,WT}}$ and $k_{\mathrm{off}}^{\mathrm{S,Mut}} = K_{M}^{\mathrm{Mut}}k_{\mathrm{on}}^{\mathrm{S,WT}} - k_{\mathrm{cat}}^{\mathrm{Mut}}$. The final estimates are shown in Table 6.

**Transitions between the active and inactive enzyme states: $k_{\mathrm{trans}}^{\mathrm{A}}$ and $k_{\mathrm{trans}}^{\mathrm{I}}$.** The ratio of the transition rate constants between the active and inactive enzymes is related to the free-energy difference $\Delta G^{\mathrm{I} \rightarrow \mathrm{A}}$ through:

$$\frac{k_{\mathrm{trans}}^{\mathrm{I}}}{k_{\mathrm{trans}}^{\mathrm{A}}} = \mathrm{e}^{-\beta \Delta G^{\mathrm{I} \rightarrow \mathrm{A}}} \ . \tag{14}$$

Note that $\Delta G^{\mathrm{I} \rightarrow \mathrm{A}}$ here is defined as $\epsilon_{\mathrm{A}} - \epsilon_{\mathrm{I}}$, as in section 3.1.2. For the mathematical, computational model this needs to be found as it is not directly measurable experimentally. Using free energy perturbation (FEP) simulations, we are able to determine the change in $\Delta G$ between each mutation and that of the WT:

$$\Delta\Delta G_{\mathrm{Mut}} = \Delta G_{\mathrm{Mut}}^{\mathrm{I} \rightarrow \mathrm{A}} - \Delta G_{\mathrm{WT}}^{\mathrm{I} \rightarrow \mathrm{A}} \ . \tag{15}$$

To find the structure of the active and inactive Abl1 WT enzyme, the X-ray crystal structures from the *Protein Data Bank* (PDB) of the Abl1 WT enzyme bound to a dasatinib molecule (PDB ID 2GQG) [45] and bound to an imatinib molecule (PDB ID 2HYY) [46] were superimposed by sequence in *BIOVIA Discovery Studio* version 2021 [47]. As dasatinib only binds to the active state of the Abl1 enzyme, this was used to find the active state. Similarly, imatinib is an inactive-binding inhibitor, so this provided the inactive state. The inactive structure had the smaller sequence and one residue missing in the middle of the chain. Due to this, the extra residues on the active structure were deleted and the missing residue was added to the inactive structure. The active state was phosphorylated in the crystal structure. To make the atomistic composition the same, the phosphate group was removed. After these changes by *BIOVIA Discovery Studio*, both models were corrected by *Swiss-Model* online tool [48].

Molecular dynamics (MD) simulations were thereafter carried out using *GROMACS* version 2021.5 [49–51]. The *Amber99sbmut* force field (modified version of *Amber99sb* used by

**Table 6. The values approximated and calculated for $k_{\mathrm{off}}^{\mathrm{S}}$ and $k_{\mathrm{on}}^{\mathrm{S}}$ for the WT and each mutant Abl1 enzyme.**

| Mutation | $k_{\mathrm{off}}^{\mathrm{S}}$ (min$^{-1}$) | $k_{\mathrm{on}}^{\mathrm{S}}$ ($\mu\mathrm{M}^{-1}\mathrm{min}^{-1}$) |
|---|---|---|
| Wild-type | 33.0 | 5.82 |
| G250E | 350 | 36.8 |
| E255K | 127 | 12.2 |
| E255V | 122 | 5.82 |
| T315I | 29.7 | 5.82 |
| T315M | 16.2 | 12.8 |
| Y253H-E255V | 26.4 | 56.6 |

*GROMACS*' plugin *pmx* to handle mutations [52, 53]) was used for the solute atoms, and the transferable intermolecular potential with 3 points model (TIP3P) was employed for water molecules [54]. A cutoff distance of 1.2 nm was used to compute van der Waals and Coulomb interactions. Long-range electrostatic interactions were modelled using the Particle Mesh Ewald (PME) method [55]. The LINCS algorithm [56, 57] was used to constrain bonds involving hydrogen atoms in the solute, and the SETTLE algorithm [58] was employed to the same aim with water molecules. The simulations were performed at constant temperature of 300 K, using the velocity rescaling thermostat [59] ($\tau_T$ = 0.1 ps), and at constant pressure of 1 bar. The pressure was kept constant by the Berendsen barostat [60] for the equilibration and by the Parrinello-Rahman barostat [61] for the transitions runs (with $\tau_P$ = 1 ps).

The *pmx* package was used for preparation of the topologies [53, 62]. Mutations (E255K, E255V, G250E, T315I, T315M and Y253H-E255V) were generated with the `pmx mutate` module. Hybrid topologies, containing information about the WT and the mutated state of the enzyme, were built from these PDB files, with the `pmx gentop` module. The system was built as a rhombic dodecahedron box, placing the enzyme at the center, at a minimal distance of 1.2 nm from each edge of the box. The protein was solvated, adding K$^+$ and Cl$^-$ ions to neutralize the charges and reach the concentration of 0.15 mM. For each mutation, a single energy minimisation was performed using a steepest descent algorithm followed by a short MD simulation of 20 ps, where positional restraints were imposed on all solute heavy atoms in order to equilibrate the water and ions around the protein. Next, the system was equilibrated for 10 ns after removing the restraints. The equilibration was repeated ten times for each state of the enzyme to ensure better accuracy through sampling. One hundred snapshots were extracted from the last 8 ns of the equilibration trajectories. From each of these snapshots, a single short MD simulation of 100 ps was spawned (200 ps for Y253H-E255V), starting from the WT (forward transition) and from the mutant (reverse transition). This resulted in simulations of 100 transitions in each direction. The Parrinello-Rahman barostat [61] was used for these simulations, and velocities were generated at each transition run. The *GROMACS* command `gmx mdrun -dhdl` was used to calculate the change in the Hamiltonian for each simulation step, where the integral along the whole path corresponds to the work done in that specific run. Finally, the average Gibbs energy for each mutation was obtained with Bennett Acceptance Ratio [63] using the `pmx analyze` module over all values and averaging over the ten simulations. The above setup was applied for both active and inactive enzymes. Of note, the enzymes were modelled without bound ATP in the simulations since it is assumed that only the active state binds ATP (Fig 1).

From the FEP simulations we can obtain the difference in Gibbs free energy upon mutation between active and inactive states ($\Delta\Delta G_{Mut}^{I \to A}$). The same information could be obtained performing two transition (inactive to active) simulations, one with the WT native protein and another after mutation. This can be seen on the closed thermodynamic cycle (Eq 16).

$$
\begin{array}{ccc}
\epsilon_I^{WT} & \xrightarrow{\ \Delta G_{WT}\ } & \epsilon_A^{WT} \\[2em]
\Big\uparrow{\scriptstyle -\Delta G_{WT \to Mut}^I} & & \Big\downarrow{\scriptstyle \Delta G_{WT \to Mut}^A} \\[2em]
\epsilon_I^{Mut} & \xleftarrow{\ -\Delta G_{Mut}\ } & \epsilon_A^{Mut}
\end{array}
\tag{16}
$$

**Table 7. The values for $\Delta G^{A}_{WT \rightarrow Mut}$ and $\Delta G^{I}_{WT \rightarrow Mut}$ with the resultant $\Delta\Delta G_{Mut}$, found through FEP simulations. $\Delta\Delta G_{Mut}$ for the wild-type is 0 kcal/mol.**

| Mutation | $\Delta G^{A}_{WT \rightarrow Mut}$ (kcal/mol) | $\Delta G^{I}_{WT \rightarrow Mut}$ (kcal/mol) | $\Delta\Delta G_{Mut}$ (kcal/mol) |
|---|---|---|---|
| G250E | −60.45 ± 0.6 | −62.66 ± 0.1 | −2.2 ± 0.6 |
| E255K | 65.60 ± 0.3 | 65.11 ± 0.5 | −0.5 ± 0.6 |
| E255V | 57.12 ± 0.2 | 56.77 ± 0.2 | −0.4 ± 0.3 |
| T315I | 29.72 ± 0.1 | 29.82 ± 0.2 | 0.1 ± 0.2 |
| T315M | 28.40 ± 0.3 | 28.50 ± 0.2 | 0.1 ± 0.3 |
| Y253H-E255V | 56.66 ± 0.4 | 57.06 ± 0.3 | 0.4 ± 0.4 |

As the clockwise sum in Eq 16 is equal to zero, we can write the following equations:

$$\Delta G^{I \rightarrow A}_{WT} + \Delta G^{A}_{WT \rightarrow Mut} - \Delta G^{I \rightarrow A}_{Mut} - \Delta G^{I}_{WT \rightarrow Mut} = 0 \; , \tag{17}$$

$$\Delta\Delta G_{Mut} = \Delta G^{A}_{WT \rightarrow Mut} - \Delta G^{I}_{WT \rightarrow Mut} \; , \tag{18}$$

and

$$\Delta G_{Mut} = \Delta G^{I \rightarrow A}_{WT} + \Delta\Delta G_{Mut} \; . \tag{19}$$

The values for $\Delta G^{A}_{WT \rightarrow Mut}$, $\Delta G^{I}_{WT \rightarrow Mut}$, and $\Delta\Delta G_{Mut}$ can be found in Table 7. It is assumed that the WT enzyme would be more stable in active state over inactive state, i.e. $\epsilon_{I} > \epsilon_{A}$, so that $\Delta G^{I \rightarrow A}_{WT} = \epsilon_{A} - \epsilon_{I} < 0$. The size of $\Delta G^{I \rightarrow A}_{WT}$ was estimated as −1 kcal/mol [37].

In order for the imatinib to be an effective drug against the WT, the inactive state must be fairly accessible. On this basis, and given $e^{-\beta \Delta G_{WT}} \approx 1$, we have assigned that $k^{A}_{trans} \approx k^{I}_{trans}$ for the WT and they are of the order of $k^{S}_{on}[S]$. This then provides the foundation for the remaining values of $k^{A,I}_{trans}$ for the mutations. As the values of $\Delta\Delta G_{Mut}$ for most of the mutations are small, with the exception of G250E, we can assume that the values for $k^{A}_{trans}$ and $k^{I}_{trans}$ for these mutations are similar to those of the WT. The sizes of these rate constants for G250E will be assumed to be slightly smaller due to its larger $\Delta\Delta G_{Mut}$. With all this information in mind, we have approximated that $k^{A}_{trans} = k^{I}_{trans} = 60 \text{ min}^{-1}$ for all mutants, with the exception of G250E in which $k^{A,I}_{trans} = 55 \text{ min}^{-1}$.

**3.1.4 Time-dependent drug concentrations in patients.** In reality, the drug concentration in the plasma of a patient is not constant, but fluctuates based on the timing between drug intake and physiological conditions. Therefore, for the time-dependent drug concentration in patients, we assume a perfect 'model' patient who receives a regular daily oral dosage of the drugs in a course of treatment with a fixed time between doses and whose body consistently maintains other biological conditions so that absorption and elimination of the drug happen in the same way every single day.

After the first dose of treatment, the concentration of the drug in the patient increases. There is a sharp absorption curve as the drug is absorbed into the bloodstream. The body begins to eliminate the drug as soon as it enters the system, however, the elimination starts off very small. At a point, the process of the drug being eliminated from the system becomes dominant and the concentration falls exponentially, usually with a long tail. The concentration continues to drop, but is still present at the point in which another dose of treatment is to be taken. With the second dose of treatment, there will be another sharp absorption curve, but this time the concentration it reaches is higher, as there is still some of the first dose remaining in the system. The remainder from the dose when the next dose is taken contributes to a taller

peak in concentration until a steady-state of concentration is reached where the dosing creates a fluctuating concentration between a consistent daily minimum and maximum.

We solve for the plasma concentration of the inhibitor $C(t)$ at time $t$ since the first dose of Abl1 inhibitor. Let $n$ be the number of days (24 hour periods) since the first dose. $k_a$ and $k_e$ are the absorption and elimination rate constants of the Abl1 inhibitor. The solution for the multiple dose pharmacokinetics for the plasma concentration of the inhibitor $C(t)$ is given by [64]

$$C(t) = \Gamma\left(\left(e^{-k_e t}\sum_{j=0}^{n}\varepsilon^j\right) - \left(e^{-k_a t}\sum_{j=0}^{n}\alpha^j\right)\right) , \tag{20}$$

where $\Gamma$, $\alpha$, and $\varepsilon$ are constants that are given by:

$$\Gamma = \frac{FDk_a}{V_d(k_a - k_e)} , \tag{21}$$

$$\alpha = e^{k_a \tau} , \tag{22}$$

and

$$\varepsilon = e^{k_e \tau} . \tag{23}$$

$F$ is the bioavailability, $D$ is the daily dose taken in moles, $V_d$ is the volume of distribution, and $\tau$ is the dosing interval. The bioavailability of imatinib is 98% [65], but only qualitative descriptions for the bioavailability of ponatinib and dasatinib were found in the literature ("high" [66] and "potent, orally bioavailable" [67], respectively). Therefore, the value for $F$ was assumed to be 1 across all three drugs. The elimination rate constant, $k_e$, was found using half-life values, $t_{1/2}^e$, as: $k_e = \ln 2/t_{1/2}^e$. Medicinal doses, $D$, are provided as a mass of the active ingredient—the molar mass was used to calculate the dose in moles. We chose a system with daily doses of treatment, hence the time between doses, $\tau$, is 24 hours. For the other values that shape the inhibitor concentration dynamics for the three drugs of interest, see Table 8.

We also used a random variance in the time a dose was taken to test the robustness of the model. The details and results of this can be found in the supplementary materials (S3 Text).

**Table 8. Values for the calculation of the time-dependent drug concentrations in the system based on observed data found and derived from a variety of sources.** Sources listed respective to imatinib, ponatinib, and dasatinib. Absorption constants, $k_a$, were taken from [68], [69], and [70]. The elimination half-lifes, $t_{1/2}^e$, that the elimination constants, $k_e$, were calculated from are from [65], [71], and [72]. For the dose, $D$, we chose masses described as daily doses in "typical treatment" [73], [69], and [74]. The volumes of distribution, $V_d$, were taken from [65], [75], and [72]. Values for $\Gamma$, $\alpha$ and $\varepsilon$ were calculated using Eqs 21–23.

| | Inhibitor drug | | |
|---|---|---|---|
| | imatinib | ponatinib | dasatinib |
| $k_a$(h$^{-1}$) | 0.94 | 1.302 | 1.74 |
| $t_{1/2}^e$ (h) | 18 | 24 | 4 |
| $k_e$(h$^{-1}$) | 0.0385 | 0.0289 | 0.173 |
| $D$ (mg) | 400 | 45 | 180 |
| $M$ (g/mol) | 493.603 | 532.6 | 488.01 |
| $D$ ($\mu$mol) | 810.4 | 84.49 | 368.8 |
| $V_d$ (L) | 435 | 1223 | 2502 |
| $\Gamma$(nM) | 2112.46 | 70.65 | 163.73 |
| $\alpha$ | 6.28E+09 | 3.72E+13 | 1.37E+18 |
| $\varepsilon$ | 2.52 | 2 | 64 |

**3.1.5 Statistical mechanics of the initial conditions of the system without treatment.**
The initial state from which the simulations start should be equivalent to the untreated condition. We use an analytical approach to determine this state. As the initial conditions are a system with no Abl1 inhibitor present, the system should reach a quasi-equilibrium steady-state. Using the results from section 3.1.2 and removing the inhibited states, we can write the steady-state distribution of a system without inhibitor as:

$$[E_I] = [E_{tot}] \frac{1}{W_{tot}^0} \ , \tag{24}$$

$$[E_A] = [E_{tot}] \frac{e^{-\beta \Delta G}}{W_{tot}^0} \ , \tag{25}$$

$$[E_A S] = [E_{tot}] \frac{e^{-\beta \Delta G_{[S]}}/K_M}{W_{tot}^0}, \tag{26}$$

where,

$$W_{tot}^0 = e^{-\beta \Delta G} \left(1 + \frac{[S]}{K_M}\right) + 1 \ . \tag{27}$$

We set the total concentration of the enzyme, $[E_{tot}]$, to 1 $\mu$M [76, 77].

## 3.2 Analytical approach to Abl1 inhibitor selection in treatment to combat drug resistance

To determine the effectiveness of treatment options on a specific mutation, in this work, we base ourselves directly on a comparison between the resulting product formation rates in our model. The maximum product formation rate, or velocity, for a given enzyme concentration is denoted by $V_{max}$, which is proportional to $k_{cat}$. The product formation rate of a system can be described in terms of $V_{max}$, $K_M$, and the substrate concentration, [S] [78].

$$\frac{d[P]}{dt} = V_{max} \frac{[S]}{[S] + K_M} \tag{28}$$

Considering just the enzyme and substrate (not treatment or other external factors), there are two main routes by which a mutation can affect this expression, either through $V_{max}$ via $k_{cat}$, or through $K_M$ as the fraction $[S]/([S] + K_M)$. As mentioned in section 2, catalytic efficiency is also a candidate for the prediction of treatment resistance. These will be presented and examined in section 4.2.

A similar equation can be derived for the product formation rate when [R] = 0 and [R] = IC$_{50}$ to derive an equation for how the product formation rate is affected by the inhibitor concentration, [R]. For this we can write the product formation rate for non-zero values of [R] in terms of the product formation rate with no inhibitor present ([R] = 0). We first consider Eq 4 and note that the inhibitor concentration only enters through the total weight $W_{tot}$, i.e.

$$\left.\frac{d[P]}{dt}\right|^{d,i,p} = \left.\frac{d[P]}{dt}\right|_{[R]=0} \frac{W_{tot}^0}{W_{tot}^{d,i,p}} \ . \tag{29}$$

From Eqs 5 and 6 we can also see that the weight is a linear function of the inhibitor

concentration (see also Fig 2). Again making use of the definition of the IC$_{50}$ through Eq 9, we find

$$\frac{\mathrm{d}[P]}{\mathrm{d}t} = \frac{\mathrm{d}[P]}{\mathrm{d}t}\bigg|_{[R]=0} \frac{\mathrm{IC}_{50}}{\mathrm{IC}_{50} + [R]} \quad . \tag{30}$$

To examine this, the fluctuating inhibitor concentrations discussed in section 3.1.4 and chosen substrate concentrations of the model will be used to illustrate whether $k_{\mathrm{cat}}$, $K_{\mathrm{M}}$, catalytic efficiency, or Eqs 28 and 30 can be used to inform the selection of Abl1 inhibitors based on present mutations.

## 4 Results

### 4.1 Computational results

We implemented the time-dependent differential equations as described in section 3.1 for the concentrations numerically using an Euler forward algorithm and therefore chose a sufficiently small time step, 1 ms.

We first consider the time-dependence of the inhibitor concentration, and then its effect on the enzyme concentrations. The inhibitor concentration for imatinib is shown in Fig 3A for the first 10 days of treatment. Each dose of imatinib taken has a short absorption curve (increase in concentration) and a longer elimination curve (decrease in concentration). Initially, there is a build-up of concentration in the system as the absorption process is much quicker than the elimination process. After a few days, the concentration enters a periodic steady-state where the concentration fluctuates between a regular maximum and minimum with every daily dose. This is also seen with the concentrations of ponatinib and dasatinib in Figs A and B in S2 Text. This variation of the inhibitor concentrations leads to variations in the concentrations of the different states of the Abl1 enzyme—an example of this effect can be seen in Fig 3C, where the states of the G250E mutation enzyme vary in response to the concentration of imatinib—a slow change towards a repeating steady-state. Further examples can be seen in Fig A in S2 Text, which shows sharper change toward the steady-state with for the combination of WT Abl1 and ponatinib; and Fig B in S2 Text, showing that dasatinib has very little influence on the states of the Abl1 which carries an T315I mutation.

As a consequence of these changes in the enzyme state concentrations, we see changes in the product formation rates, as shown in Fig 3B. These changes in product formation rate do not lead to a clear explanation of the link between product formation rate and resistance to treatment. All the mutations are expected to show resistance to imatinib, but in Fig 3B, E255V shows a product formation rate lower than the WT and T315M shows similar results to the WT. Similar discrepancies between product formation rate and resistance are seen in Figs A and B in S2 Text.

To further examine the cause of the root of drug resistance, we compare the "inhibitory reduction prowess" (IRP) of each mutation for each inhibitor. This is the difference in product formation rate in the steady-state halfway between two doses, compared with the initial product formation rate without inhibitor as a percentage of this initial product formation rate:

$$\mathrm{IRP} = \frac{\frac{\mathrm{d}[P]}{\mathrm{d}t}\big|_{[R]=0} - \frac{\mathrm{d}[P]}{\mathrm{d}t}\big|_{\mathrm{steady-state} \ +0.5\mathrm{days}}}{\frac{\mathrm{d}[P]}{\mathrm{d}t}\big|_{[R]=0}} \times 100\% \quad . \tag{31}$$

A higher value of IRP indicates that the treatment is successful at reducing the product formation rate. Since our simulations show that the steady-state is definitely reached on the 10th day for all drugs, we extract the IRP from our simulations at 9.5 days since the first dose. Fig 4

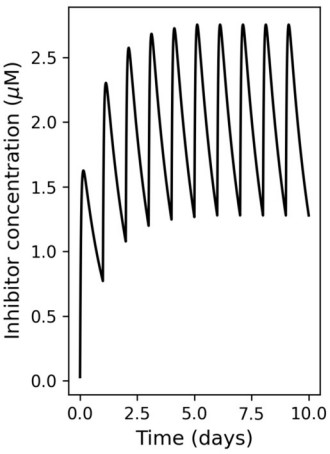

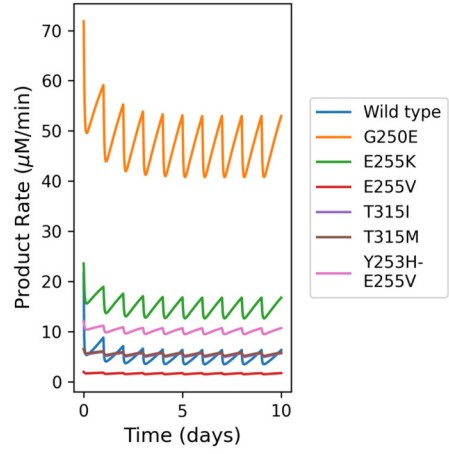

A  Concentration of imatinib in the system.

B  Product formation rates from systems with imatinib.

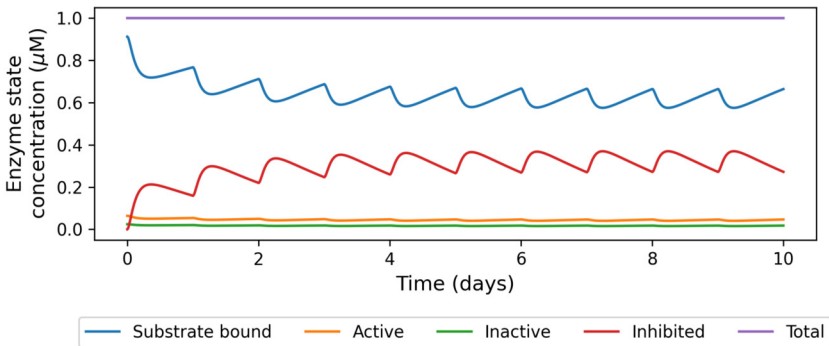

C  Effect of imatinib concentration on G250E states.

**Fig 3. Various outputs from the model with the drug imatinib.** A: Imatinib concentrations over the first 10 days of daily treatment doses, calculated as outline in section 3.1.4. The imatinib concentration increases with each daily dose (absorption), then decreases (elimination). After a build up in the system the concentration reaches a steady-state phase with a consistent minimum and maximum that the concentration fluctuates between. B: The product formation rates of the wild-type and six mutant Abl1 enzymes over ten days of initial treatment with imatinib. C: Effect of Abl1 inhibitor imatintib on the enzyme states of G250E. A gradual effect as the inhibitor concentration builds up to its steady-state and the change in substrate bound enzymes from around 0.9 $\mu$M to around an average of 0.6 $\mu$M.

shows this "inhibitory reduction prowess" and Fig 5 provides a heat map of the IRP compared with the associated resistances.

## 4.2 Choosing between treatments in the case of Abl1 inhibitor resistance using $k_{cat}$, $K_M$ and IC$_{50}$ values

We now turn to the ideas discussed in section 3.2. We compare numerical evaluations of the various indicators of resistance to what is known about resistance of the various mutation/drug combinations [14].

We first consider indicators of resistance of the mutations to treatment in general that are independent of the inhibitor: the fractional part of Eq 28 ([S]/([S] + $K_M$)), and the values of $k_{cat}$, $K_M$, and catalytic efficiency ($k_{cat}/K_M$). We also examine the information that is dependent

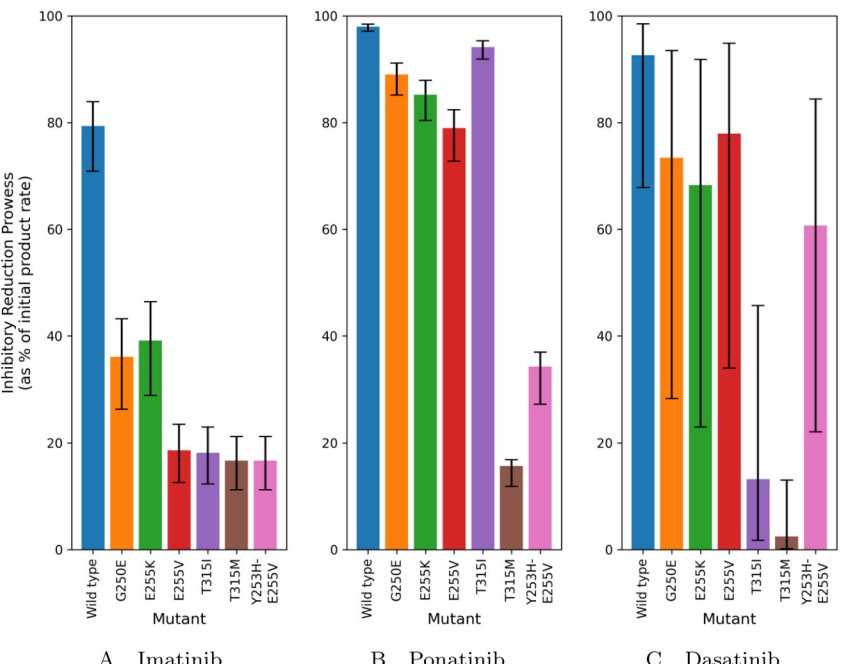

**Fig 4. We define the term "inhibitory reduction prowess" (IRP) as the percentage by which the product formation rate is reduced by from the initial point before inhibitor is introduced to the system.** A low IRP indicates a resistant mutant. The bar height is the IRP obtained from the midday level of the product formation rate on day 10 and the error bars give the range from the lowest and highest product formation rates.

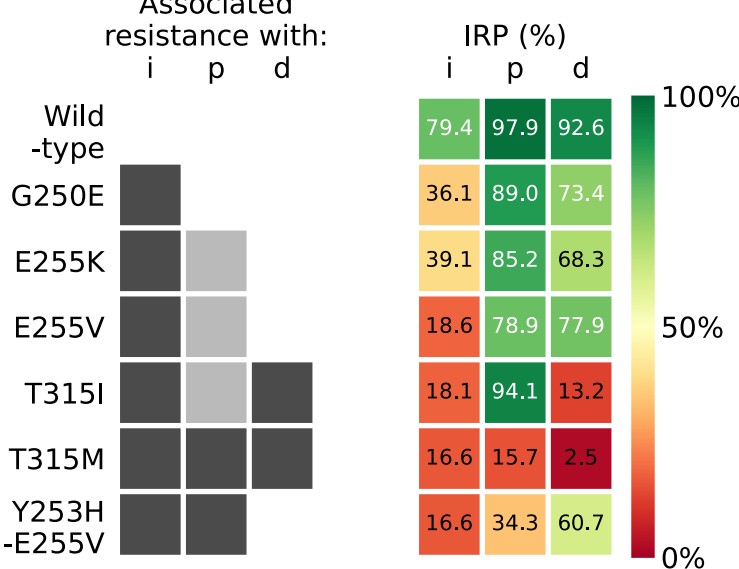

**Fig 5. A heat map to compare the associated resistances outlined in Table 1 and [14] with the IRP values.** The drugs imatinib, ponatinib, and dasatinib are represented by i, p, and d, respectively. The darker grey boxes indicate which mutations and combinations of mutations are associated with resistance to that inhibitor drug. The lighter grey boxes for ponatinib indicate that those mutations are associated with resistance when in combination with other mutations.

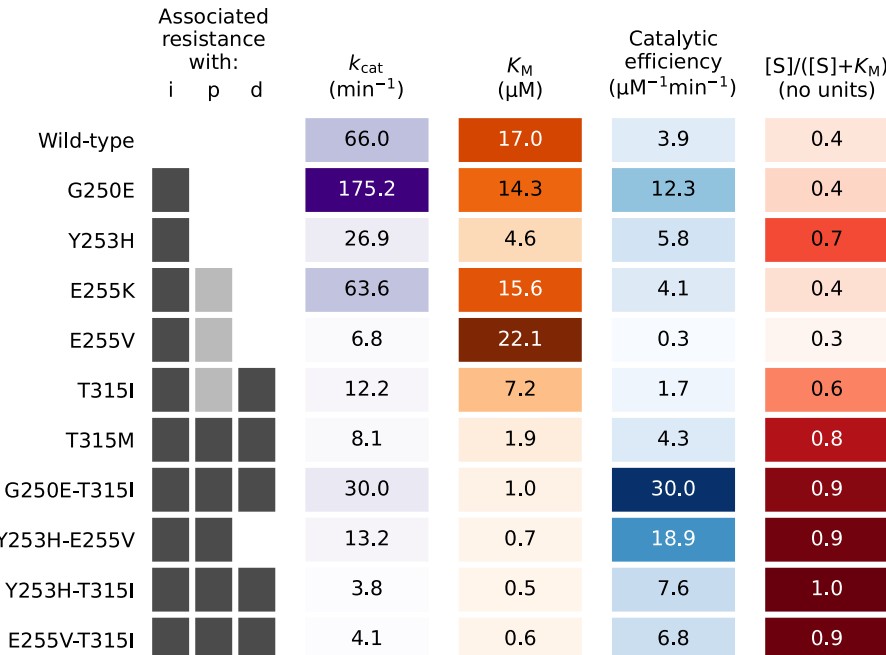

| | Associated resistance with: | | | $k_{cat}$ (min$^{-1}$) | $K_M$ (µM) | Catalytic efficiency ($\mu M^{-1} min^{-1}$) | [S]/([S]+$K_M$) (no units) |
|---|---|---|---|---|---|---|---|
| | i | p | d | | | | |
| Wild-type | | | | 66.0 | 17.0 | 3.9 | 0.4 |
| G250E | ■ | | | 175.2 | 14.3 | 12.3 | 0.4 |
| Y253H | ■ | | | 26.9 | 4.6 | 5.8 | 0.7 |
| E255K | ■ | ▨ | | 63.6 | 15.6 | 4.1 | 0.4 |
| E255V | ■ | ▨ | | 6.8 | 22.1 | 0.3 | 0.3 |
| T315I | ■ | ▨ | ■ | 12.2 | 7.2 | 1.7 | 0.6 |
| T315M | ■ | ■ | ■ | 8.1 | 1.9 | 4.3 | 0.8 |
| G250E-T315I | ■ | ■ | ■ | 30.0 | 1.0 | 30.0 | 0.9 |
| Y253H-E255V | ■ | ■ | | 13.2 | 0.7 | 18.9 | 0.9 |
| Y253H-T315I | ■ | ■ | ■ | 3.8 | 0.5 | 7.6 | 1.0 |
| E255V-T315I | ■ | ■ | ■ | 4.1 | 0.6 | 6.8 | 0.9 |

**Fig 6. A visual comparison of the associated resistance data from Table 1 with the values of $k_{cat}$, $K_M$, catalytic efficiency ($k_{cat}/K_M$), and [S]/([S] + $K_M$) for each mutation of Ab1.** The drugs imatinib, ponatinib, and dasatinib are represented by i, p, and d, respectively. For the resistance association, the darker grey boxes indicate which mutations and combinations of mutations are associated with resistance to that inhibitor drug. The lighter grey boxes for ponatinib indicate that those mutations are associated with resistance when in combination with other mutations [14]. The colour scales for $k_{cat}$, $K_M$, catalytic efficiency, and [S]/([S] + $K_M$) are simple gradient scales to emphasise the values visually with each column's smallest value as the lightest colour and the largest value as the darkest.

on the inhibitors: the current method of calculating a relative IC$_{50}$ (IC$_{50}^{Mut}$/IC$_{50}^{WT}$) and a proposed effective IC$_{50}$ ratio—the fractional part from Eq 30 (IC$_{50}$/(IC$_{50}$ + [R])).

Fig 6 shows information about associated resistance from Table 1 in comparison to values of the inhibitor independent information under investigation: $k_{cat}$, $K_M$, catalytic efficiency, and [S]/([S] + $K_M$). The association to resistance for each mutation and drug combination is shown with grey boxes—dark grey indicates an association with resistance; light grey indicates that there is association with this mutation when it is in combination with another mutation; and no colour when there is no resistance association. Although the values are listed, simple colour scales are added for $k_{cat}$, $K_M$, catalytic efficiency, and [S]/([S] + $K_M$) to emphasise the size of the values to increase clarity of their link to associated resistance. Each colour scale shows smaller values as lighter colours and larger values as darker colours.

As may be expected of values that have no dependence on the inhibitor type, there is no distinct cut-off on what determines resistance with the values examined in Fig 6. There are, however, some general trends that can be observed. Out of the three drugs we examine in this work, it appears that lower values of $k_{cat}$ and $K_M$ could indicate a higher chance of that mutation being resistant to a higher number of drugs. Conversely, the opposite trend emerges for the catalytic efficiency values. This vague correlation of number of inhibitors the mutation is resistant to and size of the catalytic efficiency seems much weaker than the correlations of $k_{cat}$ or $K_M$. The values of [S]/([S] + $K_M$) also show very limited correlation with higher values increasing the number of inhibitors the mutation will be associated with resistance to.

The values that are dependent on the inhibitor show much more promise. An examination of the fractional part of Eq 30 across a full day in steady-state of inhibitor concentration is shown in Fig 7. There is a clear divide between the resistant and non-resistant mutants with imatinib and ponatinib (Fig 7A and 7B). This divide appears to be at around IC$_{50}$/(IC$_{50}$ + [R]) = 0.5, which is where the inhibitor concentration and the IC$_{50}$ are equal. However, the dasatinib results (Fig 7C) show that this divide is not strictly obeyed. Due to dasatinib's shorter elimination half-life compared to imatinib and ponatinib, the fluctuation between the maximum and minimum points in its concentration are much larger relative to its average. The mutations that are associated with resistance to dasatinib (T315) are above this [R] = IC$_{50}$ threshold, as seen with the other two inhibitors. What is different here is that a number of the mutations that are not associated with resistance spend a significant amount of time with IC$_{50}$/(IC$_{50}$ + [R]) > 0.5, because the inhibitor concentration has decreased a lot.

Fig 8A shows the computational results compared with current methods of choosing secondary treatment when a mutation has been reported in resistant patients. Considering the fraction IC$_{50}^{\text{Mut}}$/IC$_{50}^{\text{WT}}$, for a given inhibitor, a smaller value implies that the inhibitor is a better treatment for that mutation than for a mutation where the fraction results in a higher value. In other words, it can describe which mutations that inhibitor is suited to for treatment. It does not, however, give adequate description of which inhibitor would provide the best outcome for a given mutation. The results in Fig 8A show no clear threshold in what values of IC$_{50}^{\text{Mut}}$/IC$_{50}^{\text{WT}}$ determine an outcome of resistance. Take for example the results for mutation E255K (green markers): ponatinib has the highest average IRP, but the largest value of relative IC$_{50}$. It would appear, in terms of relative IC$_{50}$, that dasatinib and imatinib would make better treatments for the E255K mutation; thus demonstrating how misleading these values are in comparing treatment options.

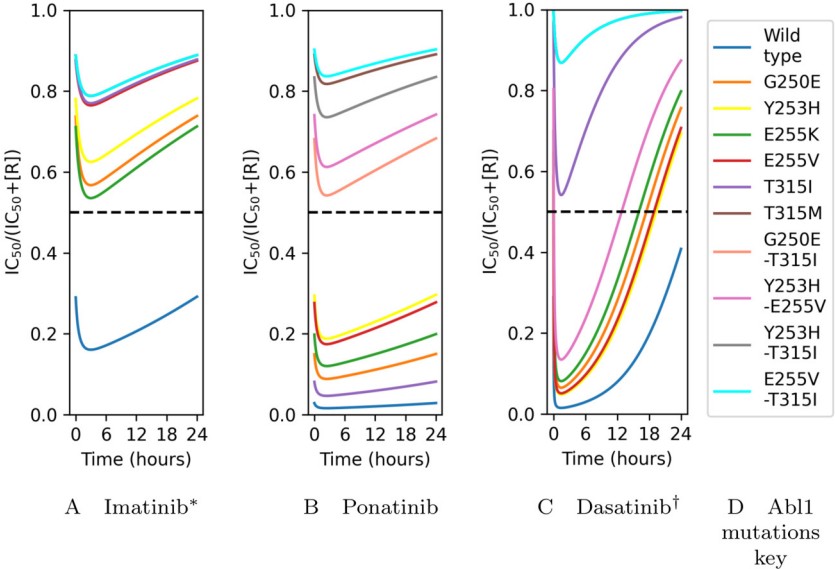

**Fig 7. How the variation in inhibitor concentration throughout one day affects the value of IC$_{50}$/(IC$_{50}$ + [R]) for different Abl1 inhibitors.** The point at which [R] = IC$_{50}$ is marked with a dashed line. A: Imatinib—G250E, E255K, E255V, T315I, T315M, and Y253H and their compound mutations are expected to show resistance. *T315M and all the compound mutations have the same IC$_{50}$ value and can be represented by the result for E255V-T315I. B: Ponatinib—Compound mutations including E255K, E255V, and T315I are expected to show resistance. C: Dasatinib—T315I and T315M are expected to show resistance. †T315M and all compound mutations containing T315I have the same IC$_{50}$ value and can be represented by the result for E255V-T315I.

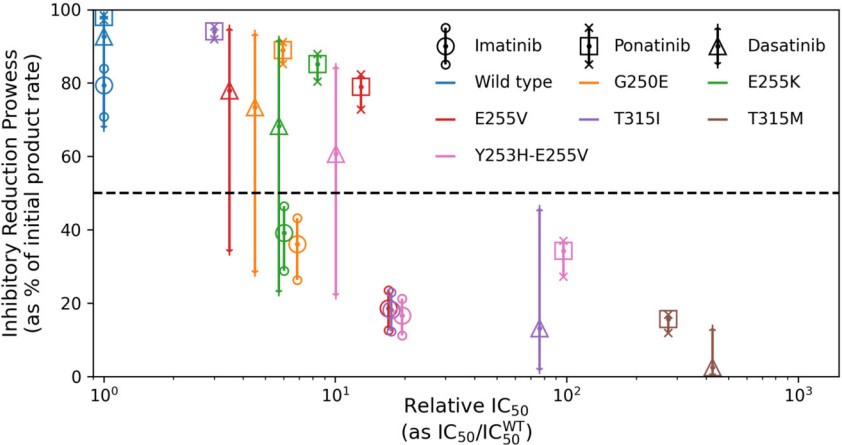

A    Relative IC$_{50}$ (as IC$_{50}$/IC$_{50}^{\text{WT}}$) against IRP.

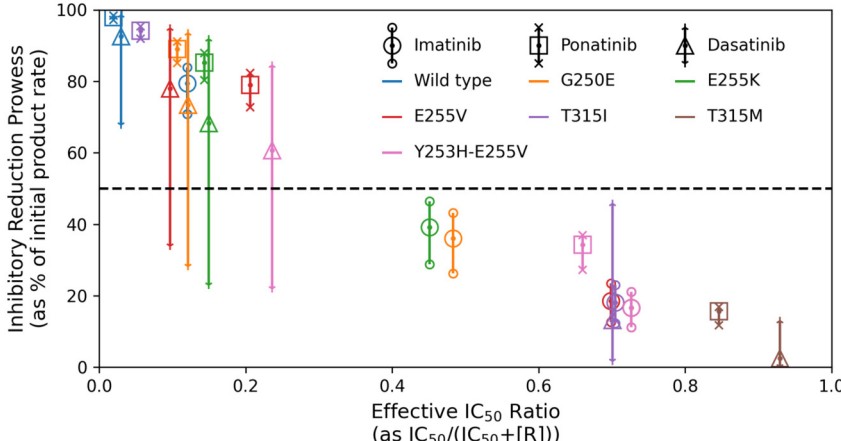

B    Effective IC$_{50}$ ratio (as IC$_{50}$/(IC$_{50}$ + [R])) against IRP.

**Fig 8. Different calculated values for indication of resistance for each combination of mutant and inhibitor against IRP.** A line where IRP = 50% shows a boundary, discussed previously, below which that combination of mutation and inhibitor displays resistance. The plotted points are the midday IRP on day 10 and the error bars give the range of the IRP for day 10. The calculated values to indicate resistance for each combination of mutant and inhibitor are A: the relative IC$_{50}$, as weighted by the IC$_{50}$ of the WT for that Abl1 inhibitor (IC$_{50}$/IC$_{50}^{\text{WT}}$); and B: the effective IC$_{50}$ ratio (as IC$_{50}$/(IC$_{50}$ + [R])).

Conversely, Fig 8B shows a much clearer divide in effective IC$_{50}$ ratio and resistance. Calculating effective IC$_{50}$ ratio as IC$_{50}$/(IC$_{50}$ + [R]) provides a way to compare between treatments for a given mutation.

## 5 Discussion

In this study, we have developed a model based on information of catalysis, pharmacokinetics and inhibition, in order to examine the effect of enzyme inhibitors when an enzyme develops resistance due to mutations in its sequence. The best indication of resistance is evident in the "inhibitory reduction prowess" (IRP—Eq 31), a normalised measure of the inhibitors

effectiveness, rather than the actual size of the product formation rate, for example see Figs 3B and 4A. This is most likely due to this system modelling patient conditions compared with measurements of IC$_{50}$ which search for a level of inhibition by varying concentration, rather than mimicking patient concentrations and measuring the effect. Further work for the model could include an examination of the behaviour of the model with asciminib in conjunction with each Abl1 inhibitor, as asciminib binds to a different site than the drugs examined in this work.

We also examined values and calculations that could be made to better select secondary treatment for identified mutations. As seen in Fig 6, the inhibitor independent values can help describe the chance of the mutation being resistant to an Abl1 inhibitor, but there is no clear correlation between these values and resistance. As evidenced in Fig 7A and 7B, resistance appears to arise when the IC$_{50}$ value is greater than the inhibitor concentration throughout the day. This is marked in these figures as IC$_{50}$/(IC$_{50}$ + [R]) = 0.5. Due to the short elimination half-life of dasatinib, the steady-state of the plasma concentration has a range that begins closer to zero than other Abl1 inhibitors. This allows for a relatively larger range of values for IC$_{50}$/(IC$_{50}$ + [R]), meaning that for a portion of each day, the system is in a "state of resistance" even for otherwise non-resistant mutants. This indicates that values of the average plasma concentrations of Abl1 inhibitors in patients combined with IC$_{50}$ information could inform oncologists when selecting which second or third generation treatment to switch to once resistance arises and the mutation has been identified. An example where the effective IC$_{50}$ ratio (IC$_{50}$/(IC$_{50}$ + [R])) paints a clearer picture of the IRP is the case of mutation E255K (indicated in green in Fig 8. E255K has IRP values of 39.1% with imatinib, 89.0% with ponatinib, and 68.3% with dasatinib, confirming its association with resistance to imatinib. The relative IC50 values (IC$_{50}^{\text{Mut}}$/IC$_{50}^{\text{WT}}$) of the mutation with each drug is 6.023 for imatinib, 7.652 for ponatinib and 5.72 for dasatinib—which gives an image that dasatinib is the best drug for treatment of this mutation, with imatinib (which is associated with resistance) in second place. The effective IC$_{50}$ ratio, aligns with the IRP results: ponatinib is the most effective course of treatment, with dasatinib in second place and resistance to imatinib.

Further comparison between Fig 8A and 8B shows that not only is there a clearer numerical divide in resistance for IC$_{50}$/(IC$_{50}$ + [R]) over the current method of IC$_{50}^{\text{Mut}}$/IC$_{50}^{\text{WT}}$, the effective IC$_{50}$ ratio occurs over a smaller, and more specific, range and has a more linear correlation to IRP. Hence, the numerical values from this effective IC$_{50}$ ratio could provide a clearer and more intuitive process for selecting secondary treatments. It would be possible to create a tool that could provide this information to oncologists and could have a great impact on patient outcomes.

As IC$_{50}$ values are used across biochemistry and pharmacology, the work here can be applicable outside of CML treatment. IRP measurements from experiments and models that mimic patient conditions could predict resistance to treatment and potentially guide treatment selection. Where this is not achievable, the inclusion of the drug concentration within a patient should be used in conjunction with IC$_{50}$ values to make a more reliable guide in treatment selection by using the effective IC$_{50}$ ratio.

## Supporting information

**S1 Text. Mass balance equations.** Table A shows the system with inactive state binding inhibitors (imatinib and ponatinib), and Table B is for system with active state binding inhibitors (dasatinib).
(PDF)

**S2 Text. Selection of results for systems with ponatinib (Fig A) and dasatinib (Fig B).**
These are similar to Fig 3 in the main text.
(PDF)

**S3 Text. A small test of the robustness of the model.** Results as IRP against time for three
drug-mutant combinations with random variance in dosing schedule as a test of robustness
for the model.
(PDF)

## Acknowledgments

The MD simulations were enabled by resources provided by LUNARC, The Centre for Scientific and Technical Computing at Lund University.

## Author Contributions

**Conceptualization:** J. Roadnight Sheehan, Astrid S. de Wijn, Ran Friedman.

**Data curation:** J. Roadnight Sheehan.

**Formal analysis:** J. Roadnight Sheehan, Thales Souza Freire.

**Investigation:** J. Roadnight Sheehan, Thales Souza Freire.

**Methodology:** J. Roadnight Sheehan, Astrid S. de Wijn, Thales Souza Freire, Ran Friedman.

**Resources:** Astrid S. de Wijn, Ran Friedman.

**Software:** J. Roadnight Sheehan.

**Supervision:** Astrid S. de Wijn, Ran Friedman.

**Visualization:** J. Roadnight Sheehan.

**Writing – original draft:** J. Roadnight Sheehan.

**Writing – review & editing:** J. Roadnight Sheehan, Astrid S. de Wijn, Ran Friedman.

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
