## [Decision Letter · Decision Letter 0]

2 Jul 2024

Dear Prof. de Wijn,

Thank you very much for submitting your manuscript "Beyond IC50 - A computational dynamic model of drug resistance in enzyme inhibition treatment." for consideration at PLOS Computational Biology.

As with all papers reviewed by the journal, your manuscript was reviewed by members of the editorial board and by several independent reviewers. In light of the reviews (below this email), we would like to invite the resubmission of a significantly-revised version that takes into account the reviewers' comments.

We cannot make any decision about publication until we have seen the revised manuscript and your response to the reviewers' comments. Your revised manuscript is also likely to be sent to reviewers for further evaluation.

Sincerely,

James Gallo

Academic Editor

PLOS Computational Biology

Jason Haugh

Section Editor

PLOS Computational Biology

Reviewer's Responses to Questions

**Comments to the Authors:**

Reviewer #1: 1) It is not evident that dasatinib cannot bind to the inactive conformation. Does the analysis change if dasatinib is treated as an active and inactive conformation binder?

2) Please clarify why the kon rate is constant for the “active” conformation binders and koff is constant for the “inactive” conformation binders.

3) G250E, E255K and E255V may favor (or bias towards) the inactive conformation (and explain why inactive conformation binders are less active) and as such the active state may not be populated as much with these mutations. Should this be taken into consideration in the weighting?

Reviewer #2: This is an interesting paper describing a numerical model of the chemical kinetics of drug resistance which is based on differential equations and a combination of known approximated and inferred rate constants.

So far as I can judge, the paper seems to be of significant interest and of good scientific quality.

Some specific comments:

(1) It would be helpful to test the numerical robustness of the system of equations to inaccuracy in the assumed rate constants and any other uncertain parameters. I would suggest applying random variations to individual parameters of a magnitude congruent with the best estimates of their uncertainty and rerunning the differential equation based numerical simulations. From this, one should be able to confirm whether robust or chaotic behaviour is observed. As many repeats as reasonably affordable should be carried out. Averaged results plus uncertainties could then be reported.

(2) On page 5, while beta is indeed the formal statistical mechanics way of naming this parameter, in practice describing its reciprocal as either kT (if using per particle energy units) or RT (if using molar units) would make the paper more accessible.

(3) On page 6, is the "total weight" the same thing as the statistical mechanical partition function? If so please state this, if not then a clear explanation of the difference would be valuable.

(4) On page 9, the general reader might infer that "dodecahedron" means the perfect Platonic solid of that name. Since this does not tessellate to fill three-dimensional Euclidean space, the authors probably have in mind some less regular dodecahedron such as the "rhombic dodecahedron" sometimes used as a simulation box where periodic boundary conditions are applied.

(5) Again on page 9, a brief explanation or justification should be provided for the assumption about the sign of the I->A free energy difference.

(6) At various points in the manuscript, in-text citations of references or equations are out of numerical order, for example "[60, 23, 32]" is seen on page 3 and "Equations 11 and 10" appears on page 8.

Reviewer #3: In this work, the authors developed a dynamic model of drug resistance to enzyme inhibitors, capturing catalysis, inhibition and pharmacokinetics, and applied it to study the effect of three Abl1 inhibitors on mutants of the Abl1 enzyme in CML. The authors use their model to argue that the relative decrease of product formation rate, which they term the “inhibitory reduction prowess,” is a better indicator of resistance than either product formation rate or fold-IC50 values for the mutant. They also discuss current practice in treatment choice and suggest a new parameter for improved efficiency of selected treatments.

The manuscript is very clearly written and easy to follow, although I found the motivation for the model and discussion of prior art someone underdeveloped and I hope the authors and bolster these sections. The model itself is quite simple to the point of being somewhat standard/trivial, but the authors collect quite a few kinetic parameters and data from the literature to integrate in the model, and the inferred metric they term IRP is an interesting deduction. My specific comments are below:

Major Comments

- The motivation for developing the model felt unclear to me after reading the introduction and section 2. Although there are other examples, I highlight three. Example 1: “Many factors have been examined in the pursuit of finding the best choice of Abl1 inhibitor to switch to once resistant mutations arise.” - This statement should have a list of citations for different methods – they only mention safety and tolerance. Example 2: “which mutations a drug is most effective against, rather than which drug is best suited to treating each mutation.” - I don’t understand the distinction. Can one not be informed by the other? Example 3: “mutations that give rise to the mutant Abl1 enzymes affect other processes in the cell.” – such as what? The authors allude to kcat not being a critical factor but it is not clear what else they are implying matters or how their model captures these other factors affected by a mutation. Are they just referring to the fact that Km might also be affected? I am not sure this qualifies as “other processes in the cell”

- It would seem to make sense to me to put the final mass balance equations for their model (e.g. the one simulated to generated Figure 4) in a supplementary table as a compact representation of the model.

- Introduction: It seems like there should be more discussion of prior art on alternative metrics to IC50 or drawbacks of IC50 e.g. https://pubmed.ncbi.nlm.nih.gov/30562549/

- Discussion: IC50s are used in many different fields – it would be nice if the authors could discuss further applications of these types of expanded dynamic models as alternatives to IC50s outside of this specific case.

- Parameter sensitivity: Is the superiority of IRP over IC50 at all dependent on the model parameters?

- At the end of the day, is this model not more explanatory than predictive, since the IRP metric was presumably developed to match the measured resistance on the left side of Figure 6? That would not be a problem in itself, but I am curious what the practical utility of the model is in terms of informing treatment decisions. For example, in the Discussion the authors suggest: “This indicates that values of the average plasma concentrations of Abl1 inhibitors in patients combined with IC50 information could inform oncologists when selecting which second or third generation treatment to switch to once resistance arises and the mutation has been identified.” Could the authors be even more specific and discuss hypothetical ranges where certain drugs would be selected under the status quo vs other drugs using the IRP-based criteria? i.e. can this model be converted into specific recommendations and potential impact assessed?

Minor Comments

- The title does not seem to reflect the focus of the Abstract – the title seems more general and the abstract more applied to a specific case. I would suggest better aligning them, maybe by instead presenting in the abstract the critical issues motivating the methodological developments in the work

- There are a lot of figures that are individually not terribly informative or are repetitive – it seems like many of them could be combined (e.g. figures 2,3,4 and 9,10).

- From Figure 1 it appears that the inhibitor binds equally well to the active and inactive form of the enzyme (uses the same parameters for each binding step). However, the authors state that “Imatinib and ponatinib both bind only to the inactive enzyme state while dasatinib binds more strongly to the active enzyme state.” – thus it appears that these parameters are actually different based on the drug and potentially mutation? It would be nice if these details were more clear from the figure, and the quoted statement should have a citation as well.

**Have the authors made all data and (if applicable) computational code underlying the findings in their manuscript fully available?**

Reviewer #1: Yes

Reviewer #2: Yes

Reviewer #3: Yes

PLOS authors have the option to publish the peer review history of their article (what does this mean?). If published, this will include your full peer review and any attached files.

Reviewer #1: No

Reviewer #2: No

Reviewer #3: No
---

## [Decision Letter · Decision Letter 1]

10 Oct 2024

Dear Prof. de Wijn,

Thank you very much for submitting your manuscript "Beyond IC50 - A computational dynamic model of drug resistance in enzyme inhibition treatment." for consideration at PLOS Computational Biology. As with all papers reviewed by the journal, your manuscript was reviewed by members of the editorial board and by several independent reviewers. The reviewers appreciated the attention to an important topic. Based on the reviews, we are likely to accept this manuscript for publication. However, one reviewer raised the following issue that we would like addressed. 

Would it have been more appropriate to have run the MD simulations in the presence of ATP? The data in the table suggests equal or more favorable kcat with most mutations which could be due to better ATP binding. The authors should at least address this question or state why this was not done (and perhaps could be looked at in the future).

Sincerely,

James Gallo

Academic Editor

PLOS Computational Biology

Jason Haugh

Section Editor

PLOS Computational Biology

Reviewer's Responses to Questions

**Comments to the Authors:**

Reviewer #1: In my earlier comments I stated in error that G250E, E255K and E255V may favor (or bias towards) the inactive conformation where I meant to say the active state may be favored. I hope this did not result in any changes.

On another front in looking at the protocol of the MD simulations to calculate free energies I am wondering if it would be more appropriate to have run these simulations in the presence of ATP. The data in table suggests equal or more favorable kcat with most mutations which could be due to better ATP binding. The authors should at least address this question or state why this was not done (and perhaps could be looked at in the future)

Reviewer #2: The authors have been helpful in their constructive responses to the review comments. I am now satisfied with their work.

Reviewer #3: The authors have sufficiently addressed my comments. I have no further issues with the manuscript

**Have the authors made all data and (if applicable) computational code underlying the findings in their manuscript fully available?**

Reviewer #1: Yes

Reviewer #2: Yes

Reviewer #3: Yes

PLOS authors have the option to publish the peer review history of their article (what does this mean?). If published, this will include your full peer review and any attached files.

Reviewer #1: No

Reviewer #2: No

Reviewer #3: No

Figure Files:

Data Requirements:

Reproducibility:

References:

---

## [Editor Report · Decision Letter 2]

18 Oct 2024

Dear Prof. de Wijn,

We are pleased to inform you that your manuscript 'Beyond IC50 - A computational dynamic model of drug resistance in enzyme inhibition treatment.' has been provisionally accepted for publication in PLOS Computational Biology.

Best regards,

James Gallo

Academic Editor

PLOS Computational Biology

Jason Haugh

Section Editor

PLOS Computational Biology

---

## [Editor Report · Acceptance letter]

1 Nov 2024

PCOMPBIOL-D-24-00563R2 

Beyond IC50 - A computational dynamic model of drug resistance in enzyme inhibition treatment.

Dear Dr de Wijn,

I am pleased to inform you that your manuscript has been formally accepted for publication in PLOS Computational Biology. Your manuscript is now with our production department and you will be notified of the publication date in due course.

With kind regards,

Anita Estes
